# Chirality amplification by desymmetrization of chiral ligand-capped nanoparticles to nanorods quantified in soft condensed matter

Ahlam Nemati[1], Sasan Shadpour[1], Lara Querciagrossa[2], Lin Li[3], Taizo Mori[4], Min Gao[1], Claudio Zannoni[2] & Torsten Hegmann [1,5]

Induction, transmission, and manipulation of chirality in molecular systems are well known, widely applied concepts. However, our understanding of how chirality of nanoscale entities can be controlled, measured, and transmitted to the environment is considerably lacking behind. Future discoveries of dynamic assemblies engineered from chiral nanomaterials, with a specific focus on shape and size effects, require exact methods to assess transmission and amplification of nanoscale chirality through space. Here we present a remarkably powerful chirality amplification approach by desymmetrization of plasmonic nanoparticles to nanorods. When bound to gold nanorods, a one order of magnitude lower number of chiral molecules induces a tighter helical distortion in the surrounding liquid crystal–a remarkable amplification of chirality through space. The change in helical distortion is consistent with a quantification of the change in overall chirality of the chiral ligand decorated nanomaterials differing in shape and size as calculated from a suitable pseudoscalar chirality indicator.

[1] Chemical Physics Interdisciplinary Program, Liquid Crystal Institute, Kent State University, Kent, OH 44242-0001, USA. [2] Dipartimento di Chimica Industriale "Toso Montanari" and INSTM, Università di Bologna, Viale Risorgimento 4, IT-40136 Bologna, Italy. [3] Institute for Smart Liquid Crystals, JITRI, Changshu 215500 Jiangsu, China. [4] Center for Materials Nanoarchitectonics (MANA), National Institute for Materials Science (NIMS), 1-1 Namiki, Tsukuba, Ibaraki 305-0044, Japan. [5] Department of Chemistry and Biochemistry, Kent State University, Kent, OH 44242-0001, USA. Correspondence and requests for materials should be addressed to T.H. (email: thegmann@kent.edu)

Homochirality—the single handedness of many key biological molecules—is ubiquitous in nature and a key signature of life on our planet. All living organisms use almost exclusively L-amino-acids and D-sugars as building blocks for proteins and nucleic acids[1,2]. Mathematically or geometrically speaking, an object is said to be chiral if it cannot be precisely mapped on its mirror image by any kind of rotation or translation. Phenomenally, chirality is observed at virtually all length scales in nature. For example, chirality or helicity in massless subatomic particles is determined by the direction of its spin and the direction of its motion (termed helicity)[3]. Here, a chiral phenomenon, rather than a chiral object, is nonsuperimposable onto its mirror image. On the other side of the scale, we are surrounded by fascinating examples of macroscopic chirality including spiral tendrils of various climbing plants[4,5], right- and left-handed (dextral and sinistral) snail shells where single gene speciation affects snail reproduction[6], and the impressive growth of exclusively left-handed up to 2.5-meter long tusks in narwhals[7,8]. The scientific community is investing a great deal of effort to elucidate the origin of homochirality, and amplification of chirality (amplified enantioselectivity) emerged as a critical underlying concept[1].

Consequently, some of the important issues we are addressing here are how far chirality reaches into the bulk from a chiral substrate, specifically a chiral nanoscale surface, and if shape or size of the nanoscale surface matter. This will also help elucidate if chirality at small scales (especially nanoscale) is amplified, and if so, can it be quantified (i.e. measured). The extent or length scale of chiral induction through space is in many cases extremely difficult or even impossible to measure. Another ubiquitous phenomenon in nature comes to the rescue; the liquid crystalline state characterized by the formation of self-assembled molecular architectures into soft condensed matter[9–12]. Liquid crystals (LCs) typically composed of low-molecular weight, rod-like organic compounds, are particularly useful to detect, measure, and even visualize chirality on several levels (atomic to macromolecular). An achiral nematic liquid crystal (N-LC) can be transformed into a cholesteric or chiral nematic LC (N*-LC) by the addition of minute amounts of a chiral solute, a so-called chiral dopant. N-LCs are one-dimensionally ordered fluids (along the director, $\hat{n}$) predominantly formed by anisometric constituents (organic or inorganic rod- or plate-like particles or molecules). Addition of a chiral solute leads to a helical distortion of the director. The helical pitch, $p$, of the induced N*-LC phase is a value that represents the extent of mutually spatial molecular interactions between host N-LC and chiral solute molecules[13]. Although, organic, molecularly dispersed systems are arguably well understood considering the large number of specifically designed and commercially available chiral molecules, arguments vary widely on why certain molecules induce tighter or larger $p$ values than others. Feringa and co-workers concluded that while no general design rules for efficient or highly potent chiral dopants could be formulated, molecular structure considerations of dopant molecules coupled to their ability to induce a certain value of $p$ often follow specific trends[14]. Recent theoretical as well as experimental work focuses especially on elucidating mechanisms that drive the formation of N*-LC phases for example using chiral hard particle or helical particle models[15,16], suspensions of rod-like viruses[17], or DNA oligomers[18]. Specific chiral additives even allow for the manipulation of chirality by light[13]. For example, using molecular motors and a shape-persistent co-dopant can create continuous rotation of N*-LC toroid-like structures able to even carry a cargo[19]. In another example, the helical pitch could be continuously tuned by near-IR illumination using a combination of a photo-isomerizable chiral dopant and upconverting nanoparticles[20].

In contrast, experimental data shedding light on the spatial extent of chiral induction from molecularly chiral nanoscale surfaces and nanoparticles are just beginning to emerge. These early experiments, however, indicate that chiral ligand-capped gold nanoparticles (Au NPs) outperform their organic molecular counterparts, inducing tighter $p$ values at lower chiral molecule concentrations[21]. In addition, these Au NPs perform this amazing feat consistently over larger distances, translating into larger chiral correlation lengths[22]. Such enhancement of through-space chirality finds support from recent examples of demonstrated long-range, through-space interactions between chiral molecules and plasmonic nanostructures[23] as well as enhanced anisotropy or Kuhn's dissymmetry factors, $g$ ($g = \Delta\varepsilon/\varepsilon$, where $\Delta\varepsilon$ and $\varepsilon$ are the molar circular dichroism and molar extinction coefficient, respectively) for chiral molecules in the vicinity of plasmonic nanostructures[24,25].

We report here that desymmetrization from nanoparticles to nanorods coupled with shape complementarity between the constituents (molecules of the surrounding medium and nanorods) translates into unparalleled high values for the helical twisting power of the admixed chiral ligand-capped gold nanorods (GNRs). A one order of magnitude lower number of chiral molecules can induce a much tighter helical distortion in a host liquid crystal medium solely by affixing chiral molecules to a nanorod instead of a quasi-spherical nanoparticle surface.

## Results

**Chirality enhancement by desymmetrization hypothesis**. To prove this hypothesis, desymmetrization of a plasmonic nanostructure by replacing a polyhedral (quasi-spherical) plasmonic NPs with a nanorod will result in further through-space chirality enhancement and even tighter $p$ values at lower overall concentrations of chiral organic molecules. Intense chiroptical signals have been reported for gold nanorods (GNRs) assembled in either helical[25] or side-by-side ladder-like fashion[26], in some case with exceptionally high $g$ factors. Hence, a combination of anisotropy and anisometry should lead to additional chirality amplification by coupling enhanced chiroptical properties with shape complementarity between an achiral structured fluid medium consisting of rod-like molecules and GNRs with a chiral organic ligand-shell. Furthermore, the admixed GNR's aspect ratio should play a critical role in this through-space chirality amplification.

This study's approach is schematically outlined in Fig. 1a, where a chiral N*-LC phase is induced by a varying concentration of GNRs capped with a virtual network of chiral cholesterol molecules using a siloxane condensation[27–29]. This approach ensured that the chiral cholesterol ligands do not desorb from the GNR surface during sonication or at elevated temperatures. A complementary set of polarized optical microscopy (POM) techniques was then used to measure $p$. At lower concentrations of any given chiral additive in N-LCs, which is before $p$ 'saturates', the inverse helical pitch $1/p$ increases linearly with chiral additive concentration, $c$. The slope of this line is then used to calculate the helical twisting power, $\beta_w$ or $\beta_{mol}$ ($\beta_w = 1/pc \cdot r$, if the weight fraction in wt.% is used for $c$, and $\beta_{mol}$ if $c$ is the molar concentration of the chiral additive with respect to the N-LC host; $r$ is the enantiomeric purity). Both are a measure of a particular chiral additive's propensity in a given N-LC host to induce a specific helical distortion of the director. We chose 4-cyano-4′-pentylbiphenyl (5CB) as N-LC host as it permitted all experiments to be performed at room temperature (Fig. 1b) and tested two sets of GNRs differing only in their aspect ratio, AR (Fig. 1c). In fact the AR of GNR1 (AR = 4.3) is roughly twice that of GNR2 (AR = 2.2).

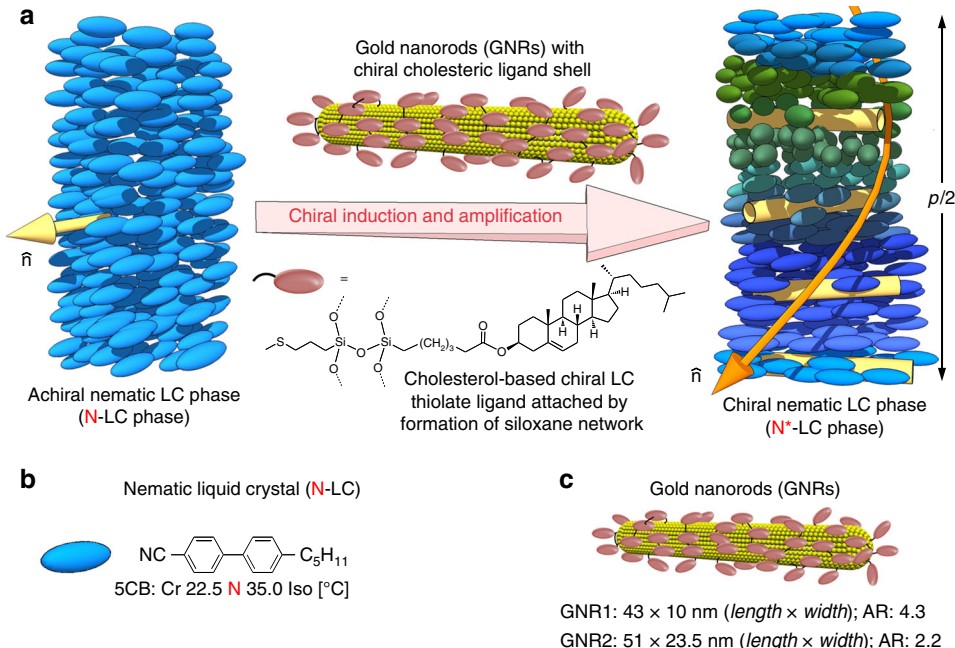

**Fig. 1** Materials and approach to testing chiral amplification. **a** Schematic representations of the N*-LC phase induced by addition of minute amounts of chiral cholesterol-capped GNRs into an achiral N-LC host. **b** The N-LC host 5CB. **c** Depiction of and values for length, width, and aspect ratio (AR) for GNR1 and GNR2

**Materials and characterization**. The ligand (Chol-silane **1**) and GNR syntheses are described in the Supplementary Information (Supplementary Note 1, Supplementary Figs. 1–6, Supplementary Table 1)[21]. Each GNR sample was fully characterized by $^1$H nuclear magnetic resonance (NMR) spectroscopy, transmission electron microscopy (TEM), vis-NIR spectrophotometry, circular dichroism (CD) spectropolarimetry, and thermogravimetric analysis (TGA) as shown in Fig. 2 and in the Supplementary Information (Supplementary Notes 2 and 3, Supplementary Figs. 7 and 8).

TEM image analysis revealed that each set of GNRs is characterized by a narrow size and shape (aspect ratio) distribution (Fig. 2a, b, Supplementary Figs. 7a and 7b). The thinner GNR1 feature an AR of 4.3 and the wider GNR2 an AR of 2.2. Each set of vis-NIR spectra showed the characteristic bathochromic shift of the transversal surface plasmon resonance (SPR) band (Fig. 2c, d) for the GNRs after ligand exchange when the cetyltrimethylammonium bromide (CTAB) ionic double layer was replaced with a (3-mercaptopropyl)trimethoxysilane (MPS)–Chol-silane **1** conjugated, thiol-based ligand shell (Supplementary Fig. 1)[29]. The solution CD spectra of the GNRs, as reported by others[30], showed no plasmonic CD signals. Spectra only showed CD signals in the UV region between 200–280 nm corresponding to the cholesterol molecules tethered at the ligand periphery (Fig. 2f). To delineate each chiral molecule's contributions in chirality transfer to the achiral N-LC host, the average number of cholesterol molecules on the GNR surface needed to be determined as precisely as possible. Figure 2g shows the data for the average weight loss determined by TGA (Supplementary Note 3, Supplementary Fig. 8) in comparison to the weight fraction of the ligand shell calculated using two separate methods as detailed in Supplementary Note 4 as well as Supplementary Tables 2 and 3. In Method 1 we used image analysis software to measure a large fraction of individual GNRs from TEM images. Taking the GNR curvature perpendicular to the long axis into account, measurements of average length, width and AR were then used to calculate the average number of Au atoms per GNR,

the number of cholesterol ligands, and the weight fraction of the ligand shell (Method 1, Supplementary Fig. 9). In Method 2 (Supplementary Fig. 10), we exclusively used a software-based approach employing MATLAB code (Supplementary Information, Appendix S1) to analyze the TEM images and calculate the average composition of the GNRs, just as in Method 1. Contrary to Method 1, however, the GNRs were modeled geometrically, composed of a flat sheet rolled-up into spherocylinders each capped with two hemispheres that is. The curvature of the spherocylinder was not taken into consideration when calculating the ligand coverage. For each method, the results obtained for the ligand shell weight fraction were then compared to the experimental TGA data (Supplementary Fig. 8 and Supplementary Tables 4 and 13). Both calculation methods are in close agreement (Fig. 2g), although the weight losses determined from the experimental TGA data are higher compared to those obtained by either calculation method. Inhomogeneities in the siloxane network ligand shell resulting in additional cholesterol ligands (Supplementary Note 5, Supplementary Fig. 11) increase the weight loss fractions in the TGA experiments that are not accounted for in each of the idealized calculations methods[31]. Nevertheless, these calculations allowed us to accurately estimate the average molecular weight of the GNRs, the number of chiral cholesterol ligands per GNR, as well as the number and the molar concentration of the GNRs and cholesterol molecules in the various, concentration-dependent mixtures in 5CB (Supplementary Note 4, Supplementary Tables 5–12, and 14–17).

**Determination of helical pitch *p***. With these data at hand, we next prepared mixtures of each of the GNRs with the N-LC host 5CB. The concentration range was varied from 0.1 to 0.7 wt.% for GNR1 and from 0.05 to 0.5 wt.% for GNR2, staying clear of higher concentrations that led to discernable GNR aggregation. At these low concentrations, the phase transition temperatures of 5CB remained essentially unaffected both on heating as well as on cooling. Differential scanning calorimetry (DSC) analysis of neat

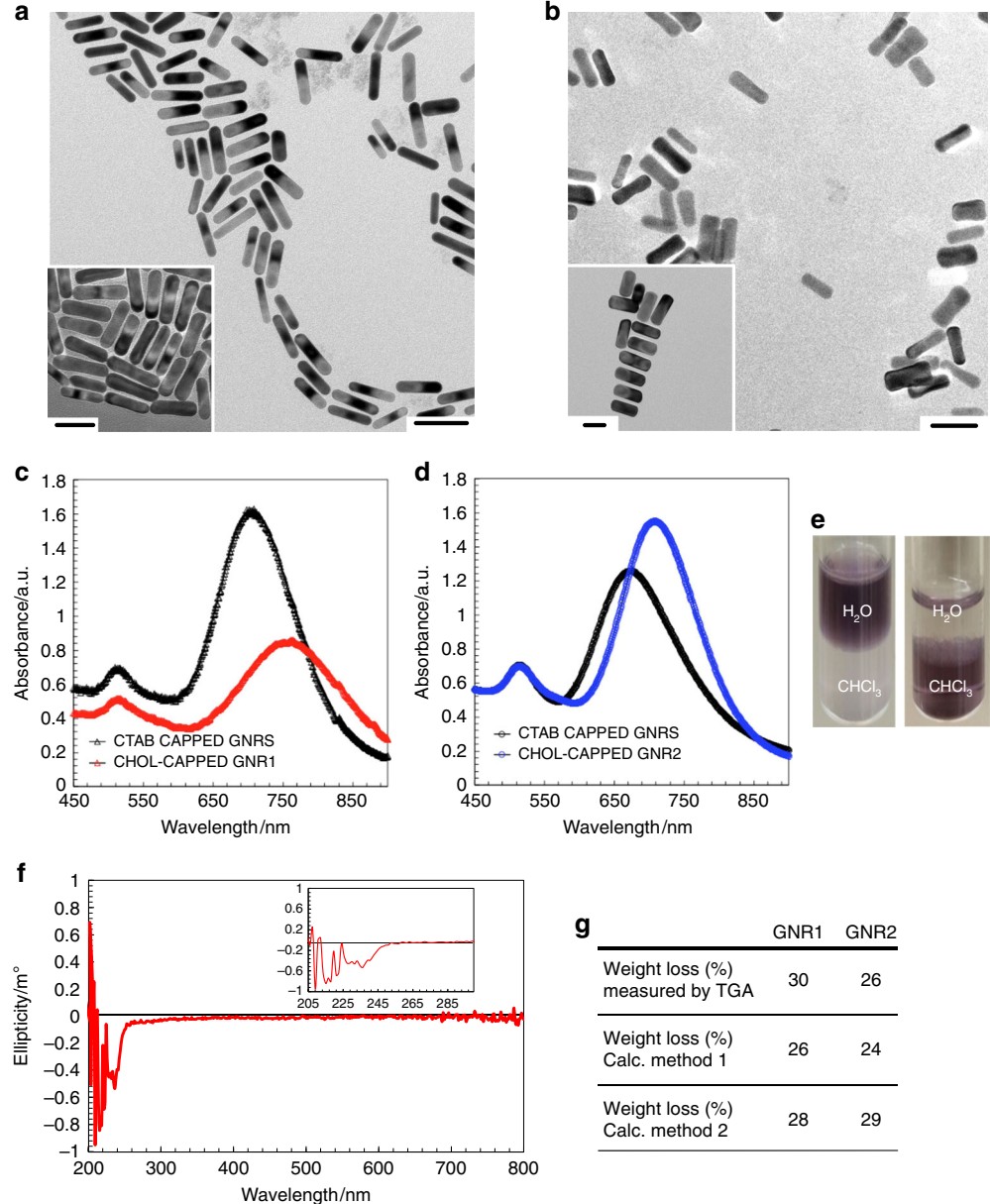

**Fig. 2** Gold nanorod characterization. TEM images of: (**a**) GNR1 and (**b**) GNR2–scale bars: 50 nm (insets show different sections of the TEM grids–scale bars: 40 nm). Vis-NIR spectra of: (**c**) GNR1 (red data points) and (**d**) GNR2 (blue data points) after siloxane condensation with the Chol-silane **1** ligand using intermediate (3-mercaptopropyl)trimethoxysilane (MPS)-coated GNRs, starting from CTAB-coated GNRs (black data points). **e** Photographs of vials of the GNRs in a biphasic CHCl₃-water mixture before phase transfer (as CTAB-coated GNRs) dispersed in the aqueous phase and after phase transfer (after siloxane condensation) in the organic phase indicated by the dark purple color. **f** Representative CD spectrum of GNR1 in cyclohexane (inset shows the 200–300 nm spectral region). The CD spectrum of GNR2 is identical. **g** Comparison of experimental (TGA) weight loss data with calculated data for the weight fraction of the organic coating (ligand shell) using two independent methods detailed in the Supplementary Information (Supplementary Note 4)

5CB and 5CB containing the GNRs (Supplementary Note 6, Supplementary Fig. 12) in fact indicated no change in phase transition temperatures as previously reported for other well-dispersed nanomaterials in N-LCs[31]. POM investigations also revealed no noticeable biphasic regions where isotropic liquid and N*-LC phase coexisted. In each case, POM textures between pre-cleaned, but otherwise untreated glass slides showed the formation of fingerprint textures indicative of the induction of a N*-LC phase after dispersing the cholesterol ligand-capped GNR1 and GNR2.

First we determined the handedness of the induced N*-LC helical molecular arrangement by preparing contact samples with a known, neat N*-LC compound, cholesteryl oleyl carbonate (COC), which forms a left-handed helical structure[22]. The lack of a discontinuation, that is, no achiral N-LC phase was induced in the contact zone, indicated that both GNRs induce a left-handed N*-LC phase (Supplementary Note 7, Supplementary Fig. 13). Induced circular dichroism spectra of the GNR/5CB mixtures confirmed the left-handedness of the induced N*-LC phase (Supplementary Note 8, Supplementary Fig. 14).

To measure p of the induced N*-LC phase, three complementary methods were used. Specifically, POM images and image analysis data were used to determine p in homeotropic (Fig. 3a-g), free surface (Fig. 3b, h–l, and Supplementary Fig. 15) and Cano

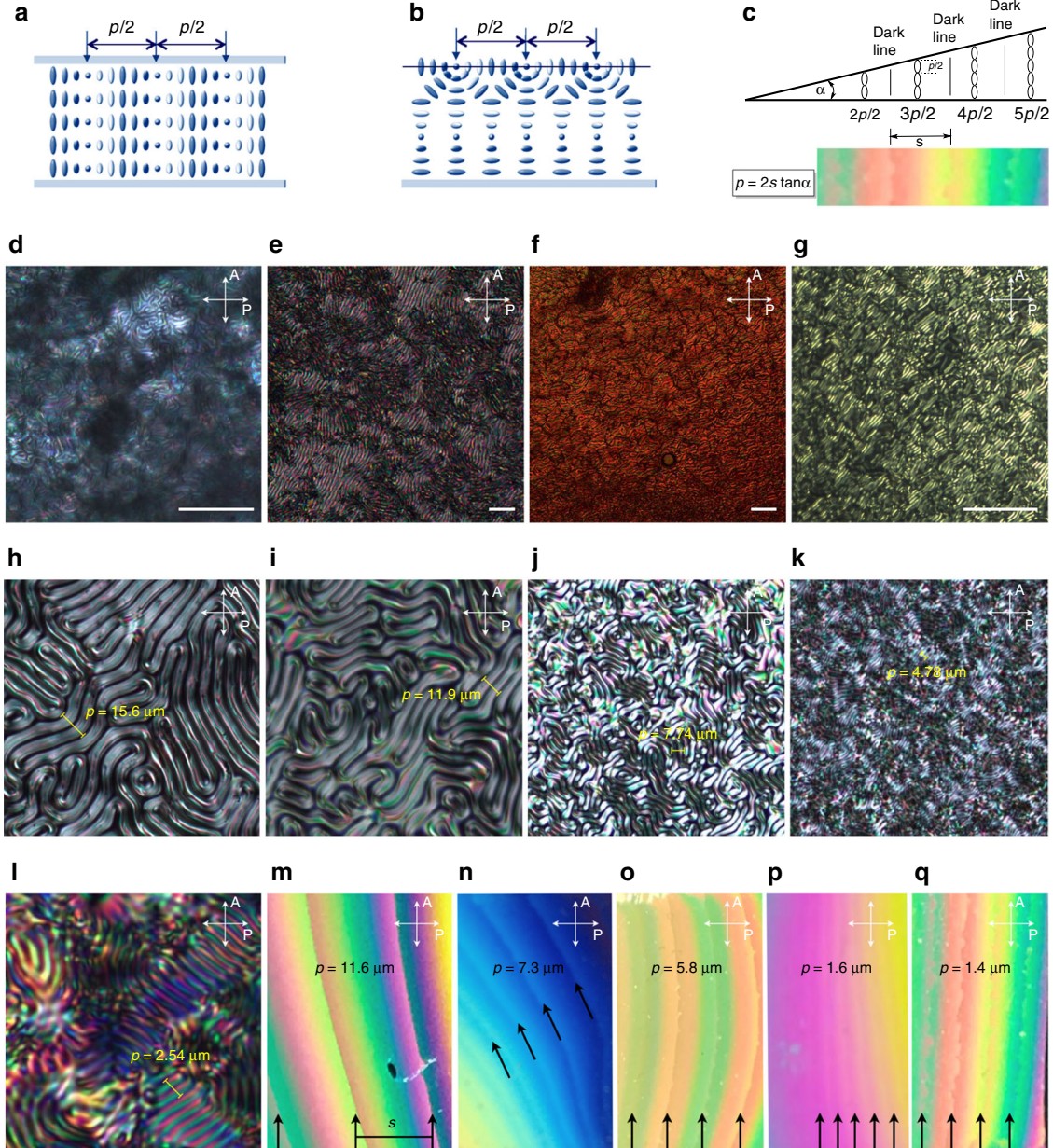

**Fig. 3** Helical pitch measurements. **a** Representation showing the orientation of the helical axis parallel to the substrate in cells treated to favor homeotropic anchoring. The half-pitch $p/2$ is measured as the distance between two dark extinction lines by POM. **b** Representation showing the spatially varying director orientation at free surfaces (bottom substrate: plain glass; top substrate: air). The determination of $p/2$ is identical. **c** Drawing schematically depicting a Cano wedge cell and an example of an N*-LC-filled wedge cell with Grandjean Cano defect lines (steps) used to measure and calculate $p$. For details see Supplementary Note 9. **d–g** POM photomicrographs (crossed polarizers) of homeotropic cells of 5CB doped with GNR1 at: (**d**) 0.1 wt.%, (**e**) 0.2 wt.%, (**f**) 0.3 wt.%, and (**g**) 0.5 wt.% (scale bars: 50 μm). **h–l** POM photomicrographs (crossed polarizers) of free surface preparations of 5CB doped with GNR2 at: (**h**) 0.05 wt.%, (**i**) 0.1 wt.%, (**j**) 0.2 wt.%, (**k**) 0.3 wt.%, and (**l**) 0.5 wt.%. **m–q** POM photomicrographs (crossed polarizers) of Cano wedge cells (with the glass surfaces modified to induce planar anchoring) of 5CB doped with GNR1 at: (**m**) 0.1 wt.%, (**n**) 0.2 wt.%, (**o**) 0.3 wt.%, (**p**) 0.4 wt.%, and (**q**) 0.5 wt.% (for more details on POM image analysis and determination of $p$, see Supplementary Note 9, Supplementary Figs. 15–26)

wedge cells (Fig. 3c, m–q, and Supplementary Figs. 18–25. Plots of the data obtained from the various cell geometry measurements for the two concentration-dependent series of GNR1 and GNR2 in 5CB showed slopes of $1/p$ vs. the GNR concentration that deviate from linearity (Supplementary Figs. 16, 17 and 26). The average of each individual slope in these plots were subsequently used to calculate $\beta_W$ and $\beta_{mol}$ considering the weight fraction (wt.%) and molar concentration of only the chiral constituents, respectively, that is the chiral cholesterol molecules

introduced by the GNRs in each mixture as calculated in Supplementary Note 4.

**Calculation and comparison of the (molar) helical twisting power.** Next we compared the ability to induce a specific value of $p$ of the two GNRs in 5CB. Moreover, we compared the GNRs' chiral induction potency in 5CB to closely related, quasi-spherical (polyhedral) cholesterol-capped Au NPs as well as to parent

organic cholesterol derivatives[21] and a range of commercially available, frequently studied organic chiral additives (Fig. 4).

The $\beta_W$ and $\beta_{mol}$ values listed in Table 1 support our initial assumption that the desymmetrization of chiral ligand-capped plasmonic nanomaterials as chiral additives amplifies chirality transfer. The values for $\beta_W$ reported here for the two GNRs are higher than some of the highest values reported in the

literature to date and the values for $\beta_{mol}$ are higher by one, even two orders of magnitude than any prior value reported for even the most potent chiral inducers of N*-LC phases. Particularly informative are the values listed for $\beta_{mol}$, since $\beta_{mol}$ is normalized to the actual number of chiral molecules present. Both $\beta_W$ and $\beta_{mol}$ increase by two orders of magnitude in comparison to the neat organic cholesterol derivatives used

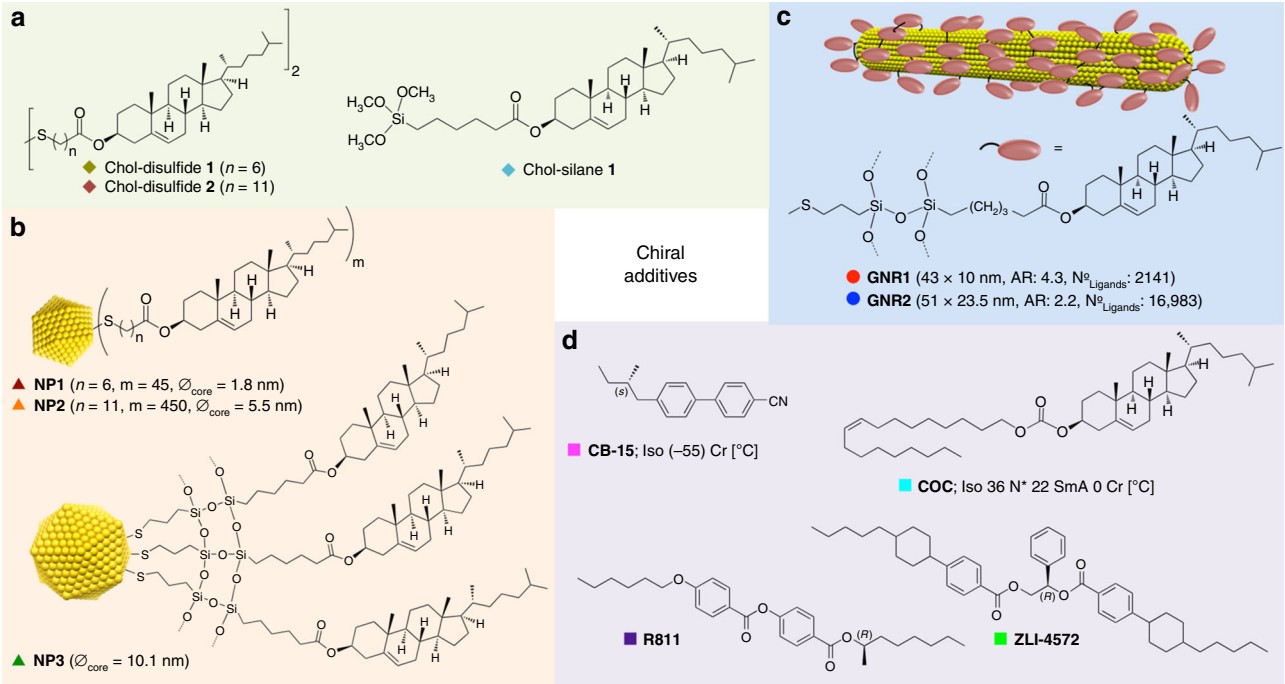

**Fig. 4** Catalogue of chiral additives. Chemical structures and additional information on the chiral additives in 5CB compared in this study (see Table 1): (**a**) chiral ligands, (**b**) chiral ligand-capped Au NPs[21], (**c**) chiral ligand-functionalized GNRs, and (**d**) commercially available chiral additives with a range of $\beta_W$ and $\beta_{mol}$ values

**Table 1 Helical twisting power ($\beta_W$), molar helical twisting power ($\beta_{mol}$), and particle-particle distances ($D_{P-P}$) for a given concentration of chiral additives in 5CB**

| Chiral additive abbreviation | $\beta_W$ / μm$^{-1}$ wt.%$^{-1}$ [a] | $\beta_{mol}$ / μm$^{-1}$ mol$^{-1}$ [a] | $D_{P-P}$ at 0.5 wt.% / nm [b] | |
|---|---|---|---|---|
| | | | center-to-center | edge-to-edge |
| Cholesterol-capped GNRs | | | | |
| GNR1 (AR: 4.3) | −382 ± 94[c] | −5723 ± 1506[c] | 54 | 39 |
| GNR2 (AR: 2.2) | −248 ± 58[c] | −3727 ± 568[c] | 93 | 66 |
| Cholesterol-capped Au NPs | | | | |
| NP1 (∅ = 1.8 nm)[21] | −8.7 | −42.2 | 9.7 | 3.0 |
| NP2 (∅ = 5.5 nm)[21] | −29.0 | −319.8 | 35 | 24 |
| NP3 (∅ = 10.1 nm)[21] | →0[d] | →0[d] | 68 | 53 |
| Cholesterol-based ligands | | | | |
| Chol-silane **1** | −2.8 | −6.0 | 3.7 | 3.25 |
| Chol-disulfide **1**[21] | −5.0 | −19.0 | 3.35 | 2.9 |
| Chol-disulfide **2**[21] | −5.0 | −22.0 | 3.5 | 3.0 |
| Commercially available and frequently used organic chiral additives | | | | |
| COC | −7.4 | −40.4 | 3.7 | 3.2 |
| CB-15[55] | +8.8 | +21.3 | 2.9 | 2.5 |
| R811[55] | +12.1 | +44.0 | 3.2 | 2.8 |
| ZLI-4572 | +30.3 | +157.7 | 3.6 | 3.2 |

[a]The sign indicates the handedness of the induced N*-LC phase. The opposite enantiomer of a particular chiral additive molecule gives the same value but opposite sign; a negative sign indicates a left-handed N*-LC phase and a positive sign a right-handed
[b]The particle-particle distance, $D_{P-P}$ (center-to-center and edge-to-edge), refers to the chiral correlation length, that is the distance between chiral additive particles or molecules to maintain a certain helical pitch throughout the bulk
[c]The particular ranges of these values were derived from the p values measured at lower concentrations (up to 0.3 wt.%) and the p values after cooperative amplification at higher concentrations of the GNRs in 5CB (>0.3 wt.%) (see Fig. 6h)
[d]The helical pitch was too large to be measured, and the sample appeared achiral nematic as indicated by the appearance of a typical Schlieren texture. Hence, no HTP value could be calculated

in the surface functionalization of the Au NPs, and by one or two orders of magnitude when compared to the actual cholesterol-capped Au NPs synthesized in the presence of the cholesterol disulfides. The presence of the chiral surface ligands inferred a chiral bias during the formation of NP1 and NP2 rendering the Au NP cores chiral as indicated by plasmonic CD signals in solution. The largest of the three Au NPs, NP3, has the exact same surface chemistry as the two GNRs described here (siloxane-conjugated cholesterol outer shell). However, $\beta_W$ and $\beta_{mol}$ could not be calculated since NP3 (core diameter $\varnothing = 10$ nm[21] identical to the width of GNR1) induced a helical pitch that was too large to be identified or measured by any of the techniques (cell geometries) described here. Noteworthy is that NP3 was not synthesized in the presence of a chiral bias. A comparison of NP3 with GNR1 is therefore an excellent example for the amplification of chirality transfer by desymmetrization as both plasmonic nanomaterials feature the exact same surface chemistry, the same ligand corona, and the same diameter. The key difference is shape anisotropy that leads to $\beta_{mol}$ values that increase from approaching zero for NP3 to an absolute value well above 5,700 $\mu m^{-1} mol^{-1}$ for GNR1. The same comparison can be made between GNR1 and GNR2. As hypothesized, GNR1 shows larger values for $\beta_W$ and $\beta_{mol}$ in 5CB than GNR2 solely due to its larger AR. Both GNR1 and GNR2 also show significantly higher values for $\beta_W$ and $\beta_{mol}$ than the cholesterol thiolate-capped NP1 as well as NP2. Both Au NPs showed plasmonic CD bands, indicating chiral NP core or surface structures, and higher $\beta_W$ and $\beta_{mol}$ values than their chiral ligands (the cholesterol disulfides **1** and **2**) that generated the in situ chiral bias during the NP synthesis[21]. Despite contributions from the chiral Au NP cores or surfaces, the GNRs proved to be more potent chiral inducers with one-to-two orders of magnitude higher values for $\beta_W$ and $\beta_{mol}$.

The GNRs also outperform several commercially available and frequently studied chiral dopants (COC, CB-15, R811, and ZLI-4572). Even the most powerful in this series, ZLI-4572, could just about compete with **NP2** (similar $\beta_W$ but 1/2 $\beta_{mol}$), but not with GNR1 or GNR2 (Table 1, Supplementary Note 10, Supplementary Figs. 27 and 28).

The two plots shown in Fig. 5 visualize the data listed in Table 1, plotting $\beta_{mol}$ vs. mol% of the various chiral additives at a given percentage by weight (0.5 wt.% exactly). Figure 5a provides the data points with annotations grouped by the type of chiral additive (squares and diamonds for neat organic chiral additives, triangles for the cholesterol-capped Au NPs and circles for the cholesterolsiloxane-capped GNRs). Figure 5b more clearly shows what specific contributions lead to the observation that the three distinct types of chiral additives (neat organic molecules, Au NPs, and GNRs) somewhat occupy specific corners within this plot. Considering the chirality transfer efficiency to the surrounding N-LC medium, chiral additives that occupy the top-right corner of this plot are those that significantly amplify chirality transfer through space and are therefore characterized by an increasing chiral correlation or persistence length (see values in Table 1). In other words, chiral additives in that corner of the plot are capable of inducing a much tighter $p$ in the bulk of the N-LC host medium at a steadily declining molar concentration of the chiral molecular constituents vis-à-vis an increasing distance between them.

**Chiral correlation or persistence length.** The calculation of this quantity, the chiral correlation or persistence length, signified by the average chiral additive particle-to-particle distance ($D_{P-P}$) that maintains a constant $p$ throughout the bulk LC phase, is detailed in Supplementary Note 11 and Supplementary Fig. 29, and the values are listed together with $\beta_W$ and $\beta_{mol}$ in Table 1. These values show the contributions of chirality transfer amplification

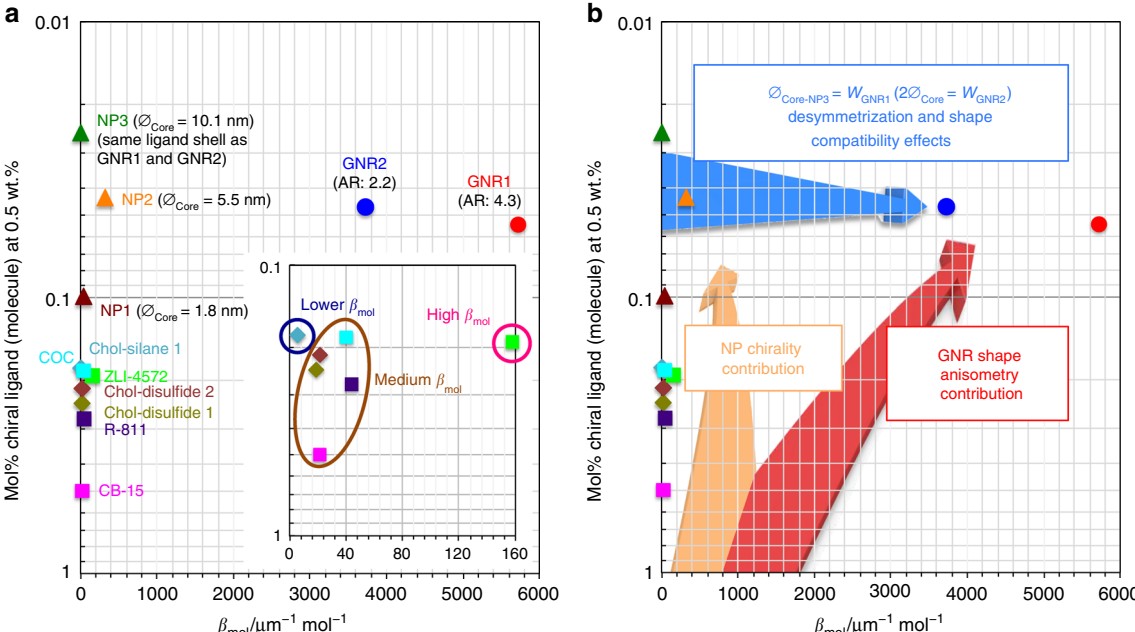

**Fig. 5** Chirality transfer efficiency. **a** Plot of the molar helical twisting power $\beta_{mol}$ ($\mu m^{-1} mol^{-1}$) vs. the concentration (mol fraction, mol%) calculated for 0.5 wt.% of the chiral additive. Diamonds and squares represent the neat organic chiral additives (blue diamonds: Chol-silane **1**, dark green diamonds: Chol-disulfide **1**, brown diamonds: Chol-disulfide **2**, pink squares: CB-15, light blue squares: COC, green squares: ZLI-4572, and purple squares: R811), triangles the chiral ligand-capped Au NPs (dark brown triangles: NP1, orange triangles: NP2, and green triangles: NP3), and circles the two cholesterol-functionalized GNRs (red circles: GNR1 and blue circles: GNR2). **b** The same plot highlighting the specific contributions of inherent NP chirality, anisometry of the GNRs and desymmetrization from chiral ligand-capped Au NPs to GNRs. Chiral correlation or persistence lengths are the lowest at the origin and highest in the top-right corner of this plot

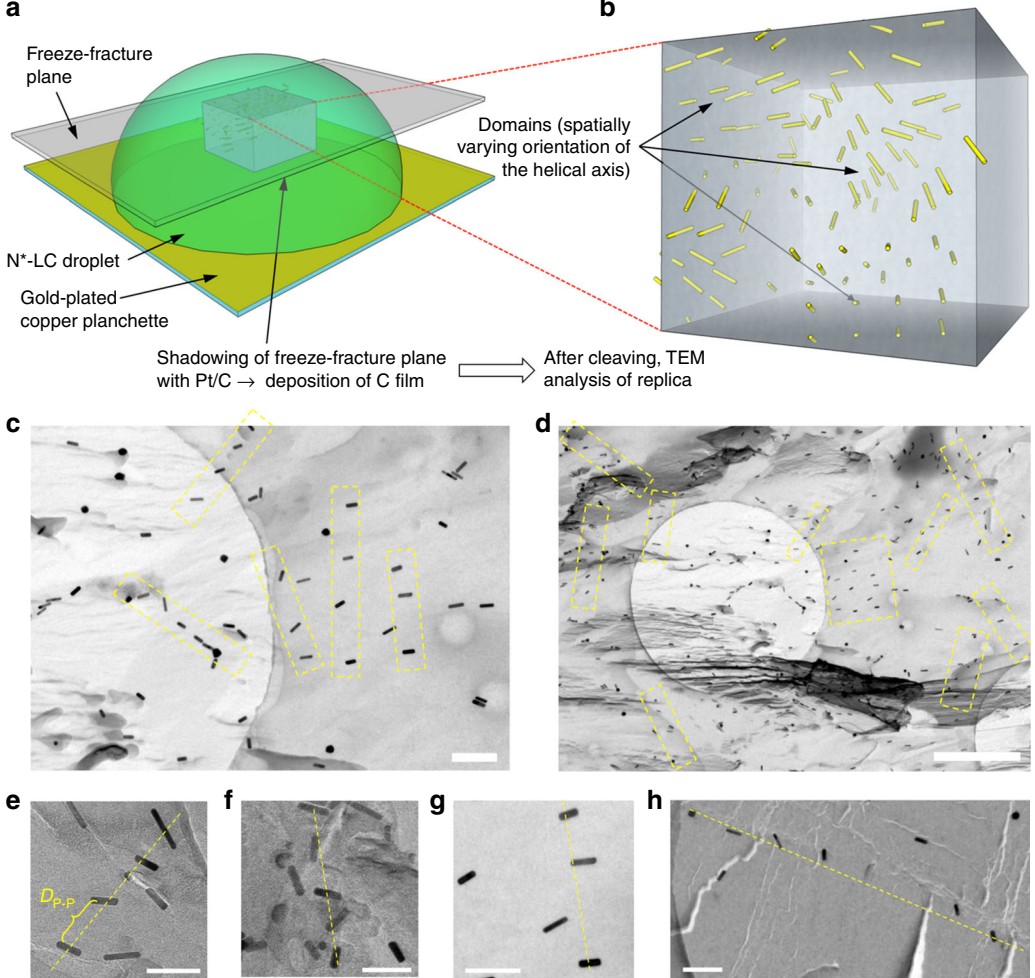

**Fig. 6** Freeze-fracture transmission electron microscopy images of induced N*-LC droplets. **a** Schematic depiction of the freeze-fracture method to prepare the TEM specimen. **b** A representative voxel of the droplet showing the multi-domain structure via the embedded GNRs. **c**, **d** FF-TEM images of the induced N*-LC phase of Felix-2900–03 containing 0.5 wt.% GNR2 (scale bars: **c** 200 nm and **d** 1 μm, apparent circles are from the TEM grids). To obtain these images, it was important that the replica captured most of the GNRs on the fractured surface, thereby providing a direct visualization of the GNRs in the bulk material. Areas highlighted by a yellow box and **e**–**h** select areas from many of the obtained FF-TEM images show GNR arrays (often twisted) with an average separation close to the calculated $D_{P-P}$ values (scale bars in **e**–**h**: 100 nm)

through space, and together with the values for $\beta_{mol}$ demonstrate that affixing an otherwise 'weaker' chiral inducer such as cholesterol to the surface of an anisometric plasmonic nanostructure can lead to substantial chiral amplification through space, here in the bulk of an N-LC. At least one order of magnitude fewer chiral cholesterol molecules are required to induce values of $p$ that are considerably smaller (tighter pitch) at significantly larger average spatial separations between the chiral additives dispersed in the LC bulk. Powerful examples to support this claim are comparisons between neat cholesterol derivatives such as Chol-silane **1**, the Chol-disulfides **1** and **2** as well as COC *vs.* the GNRs as additives in 5CB at 0.5 wt.% each. $D_{P-P}$ significantly increases as we compare the neat, free cholesterol-derivatives with the cholesterol-capped **NP1** and **NP2** inducing a measurable helical pitch (see footnote "b" in Table 1). Values for $D_{P-P}$ similar to the free cholesterol derivatives (ligands) were, as expected, also obtained for a range of commercially available chiral dopants.

To verify that the tremendous amplification of chirality through space is not unique to one specific N-LC host (here 5CB), we also measured $p$ depending on the GNR concentration in another N-LC host, Felix-2900–03, at the same reduced temperature $T_{Iso \to N} - T = 15\,^{\circ}\mathrm{C}$. The data shown in

Supplementary Note 12 (Supplementary Fig. 30) clearly demonstrate that this effect is universal and reproducible in structurally unrelated N-LC host materials.

While we do not assume that the GNRs are uniformly (spatially) distributed in the induced N*-LC matrix, considerations of local anchoring of host molecules on the GNR surface, orientational flexibility of the cholesterol molecules on the GNR surface, and the Frank elastic constants of the host N*-LC medium, significant reorientations of the GNRs away from local director ($\hat{n}$) orientations can be considered negligible. Interactions between the GNRs and the elastic N*-LC medium are sufficiently strong to overcome interparticle interactions between the GNRs that would lead to appreciable GNR aggregation[32]. Some aggregation in defects, either end-to-end[27] or side-by-side cannot be fully disqualified, but the bulk induction of practically identical $p$ values in various cell geometries and ensuing director field orientations would argue against significant aggregation of the GNRs facilitated by elasticity-mediated interaction forces. Several recent examples of various types of nanorods in N-LCs appear to support these assumptions especially if the surface chemistry, which governs the type and strength of anchoring at the interface between the GNR ligand shell and the surrounding

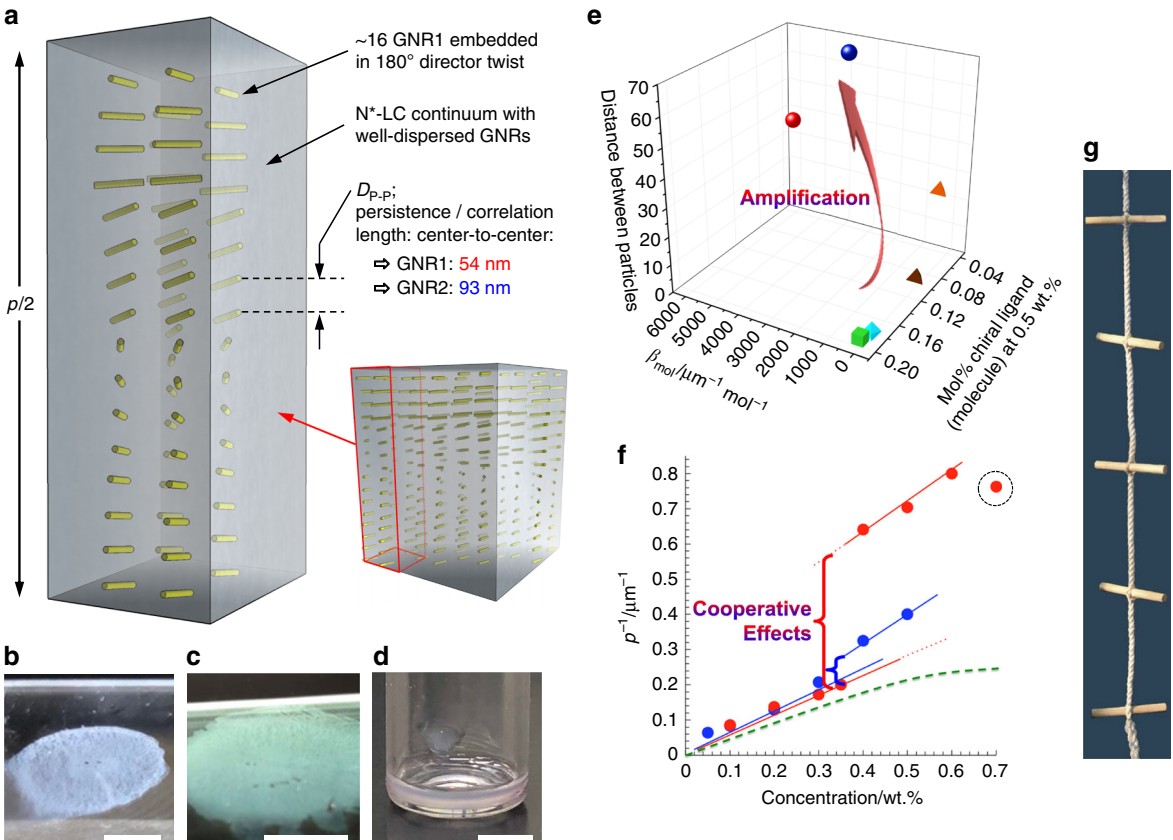

**Fig. 7** Schematic illustration of the persistence (correlation) length. **a** Idealized distribution of the cholesterol-capped GNRs in the induced N*-LC bulk assuming planar boundary conditions: the interparticle distances given are for 0.5 wt.% GNR1 in 5CB with a helical pitch $p = 1.5$ µm; $D_{P-P}$ values were calculated for both GNR1 and GNR2. **b**, **c** Photographs showing blue or green iridescent (reflection) colors depending on the viewing angle of thin films of 5CB containing 0.6 wt.% of GNR1 between polyimide-coated glass slides promoting planar anchoring (cell gap: 10 µm, scale bar: 0.5 cm). **d** Photograph of glass vial containing 5CB doped with 0.5 wt.% GNR1 showing iridescence due to wavelength- and angle-dependent Bragg-type reflections (scale bar: 0.5 cm). **e** 3D chirality transfer efficiency plot: ($\beta_{mol}$ vs. concentration) vs. $D_{P-P}$ (for symbol legend see caption Fig. 5). **f** Plot of the inverse helical pitch $1/p$ (µm$^{-1}$) vs. the concentration (wt.%) of GNR1 (red) and GNR2 (blue) in 5CB. A cooperative effect between neighboring GNRs, once a certain minimum distance between them is reached, causes a sudden substantial increase (jump) in the $1/p$ data. The hypothetical dashed green line shows the trend generally observed for organic chiral dopants–a linear relationship at lower concentrations of the chiral additive that plateaus once a certain concentration is reached, i.e., the helical pitch $p$ begins to saturate at this point. The sudden drop in $1/p$ value at a concentration of 0.7 wt.% of GNR1 in 5CB suggests the onset of GNR aggregation (dashed circle). **g** Photograph of a firecracker ladder: the steps representing the chiral ligand-capped GNRs inducing the helical twist in the rope

N-LC host molecules[33], is adjusted to maximize dispersion in the N-LC host[34–36].

Freeze-fracture TEM (FF-TEM) images obtained from a droplet of the induced N*-LC phase on glass (Fig. 6a, b) clearly support the calculated $D_{P-P}$ values, here for the mixture of 0.5 wt.% of the GNR2 in Felix-2900–03, showing both a twist among the GNRs as well as the spatial separation in many areas of these FF-TEM images of about 95 nm calculated a priori using pure geometrical data (Fig. 6c–h). As shown in Fig. 6, in the absence of well-defined boundary conditions (here glass at the bottom and air at the top) these droplets form multi-domain N*-LC structures, which are visualized in many areas via the GNRs captured by the platinum film replica. Additional FF-TEM images are collected in Supplementary Note 13, Supplementary Fig. 31a–f. To further validate the FF-TEM technique used for the GNR samples we produced comparable FF-TEM specimen for an N*-LC phase induced by one of the NPs (0.5 wt.% NP2 in Felix-2900–03; Supplementary Note 13, Supplementary Fig. 31g, h). Again, these images support the claim of well-dispersed NPs

and show several areas with average NP separations matching the calculated $D_{P-P}$ value.

An idealized representation of the distribution of the GNRs in the induced N*-LC bulk assuming well-defined planar boundary conditions is shown in Fig. 7a. The spatial separation indicated between the GNRs is based on the calculated interparticle distance, $D_{P-P}$, listed in Table 1 and on the distance measured in many areas of the FF-TEM images. Highly significant is the much larger distance between the chiral GNR building blocks inducing the bulk N*-LC helical arrangement with a spatially constant helical pitch compared to any other chiral additive reported to date. Mixtures of **GNR1** or **GNR2** in 5CB at 0.5 wt.% featuring sub-mM concentrations of cholesterol molecules, respectively, induce tight enough $p$ values for the induced N*-LC phases to show iridescence as a result of wavelength- and angle-dependent Bragg-type reflections (Fig. 7b–d). The plot in Fig. 7e clearly reveals this apparent amplification of chirality transfer by the GNRs. Cooperative effects appear to drive the deviation from linearity in the $1/p$ vs. concentration plots at concentrations of the GNRs in the N-LC hosts below the onset of aggregation > 0.6 wt.% for GNR1 and >0.5

wt.% for GNR2. Instead of showing the generally observed plateau at higher concentrations of the chiral additive, because $p$ eventually saturates, both GNR1 and GNR2 in 5CB furnish a distinct jump in the $1/p$ data as their concentrations increase (Fig. 7f).

Considering the aforementioned arguments of the influence of the GNRs' AR, such cooperative effects would have to be more pronounced for the GNR with a higher AR. This is exactly what was found experimentally for GNR1, whose $D_{P-P}$ values are consistently lower than those calculated for GNR2 (red dataset in Fig. 7f). Such cooperative contributions would explain the trend observed for $p$ as the GNR concentration increases. The chiral ligand-capped GNRs, once incorporated into the helically distorted elastic medium, create a chiral feedback loop that reinforces the helical distortion[37]. Much like in a firecracker ladder (Fig. 7g), where the rope can be twisted much easier when a torque is applied to the steps instead of to the rope itself. An additional contribution to the significant chirality amplification observed here experimentally could be the dipole-like electrostatic asymmetry of the gold nanorods themselves. Using a range of electron microscopy techniques, Kelvin probe microscopy, and simulations Kotov and co-workers showed that gold nanorods capped with CTAB surfactant bilayers (the starting point for the GNR synthesis and surface modification described herein) are non-centrosymmetric resulting from an uneven distribution of CTAB molecules on the nanorod surface[38].

**Quantification of capped nanoparticles chirality.** To gain further insight we have also proceeded to calculating a chirality indicator for the various ligand-decorated GNRs and quasi-spherical NPs.

A number of different metrics have been proposed in the literature attempting to quantify the chirality of a molecule or a certain object. Some of these are based on the distance between a given structure and the closest achiral structure[39] or on the Hausdorff distance between the sets of points representing two specular images[40]. Here we employ another approach that introduces an absolute, rather than relative to a reference, pseudoscalar indicator derived from the molecule (or more generally object) geometry[41–43]. In particular, we have estimated the overall chirality of both rod-like (GNRs) and spherical[21]

(Au NPs) nanomaterial shapes, capped with the chiral cholesterol ligands, with the help of the chirality index $G_{oa}^a$ that we have previously validated and employed for small molecules[44] and proteins[43] but also, recently, for problems as diverse as establishing the chirality of erodium awns[45]. This index depends only on geometric information, i.e. in this case the position and orientation of the ligands with respect to the nanoparticle frame, and thus indirectly on the shape and size of the nanomaterial:

$$G_{oa}^a = \sum_P \begin{cases} \frac{[(\mathbf{r}_{ij} \times \mathbf{r}_{kl}) \cdot \mathbf{r}_{il}](\mathbf{r}_{ij} \cdot \mathbf{r}_{jk})(\mathbf{r}_{jk} \cdot \mathbf{r}_{kl})}{N(\mathbf{r}_{ij}\mathbf{r}_{jk}\mathbf{r}_{kl})^2\mathbf{r}_{il}} & \text{if } i<j<k<l \in [1, \ldots, N] \\ 0 & \text{otherwise} \end{cases} \quad (1)$$

where $N$ is the total number of atoms (or beads) and $\mathbf{r}_{ij}$ is the vector distance between any two atoms i and j. We have employed $G_{oa}^a$ to examine if changing the number, position and orientation of the ligands on the surface of nanomaterials differing in size and shape can lead to variations in chirality similar to the aforementioned experiments, with the usual assumption that the helical twisting power is proportional to the chirality. To do this we have adopted a Coarse Grained (CG) representation of the Chol-silane ligands (Fig. 8a, b) and we have distributed a certain number of these ligands on the surface of the GNRs and the Au NPs with relative dimensions corresponding to the experimental data (Fig. 8c, d).

We consider the chirality of an individual GNR or NP capped with a certain number of ligands at a time, i.e. we assume that the NP interactions with each other as well aggregation of the GNRs and NPs are negligible. Next, we choose an increasing number of ligands, even if much smaller than experimental numbers (the maximum is of only 20 Chol-silane ligand molecules). We then choose a certain number of different positions and orientations of CG-ligands for a given GNR or NP, as detailed in the Supplementary Note 14, calculate their chirality with Eq. 1 and take an average in order to obtain an overall resulting $|\langle G_{oa}^a \rangle|$. To compare these results with the experimental data of Table 1, we have converted the values of helical twisting power $\beta$ into percentages. Since the trends of $\beta_W$ and $\beta_{mol}$ are quite similar, these two quantities have also been averaged. In Fig. 9, we have

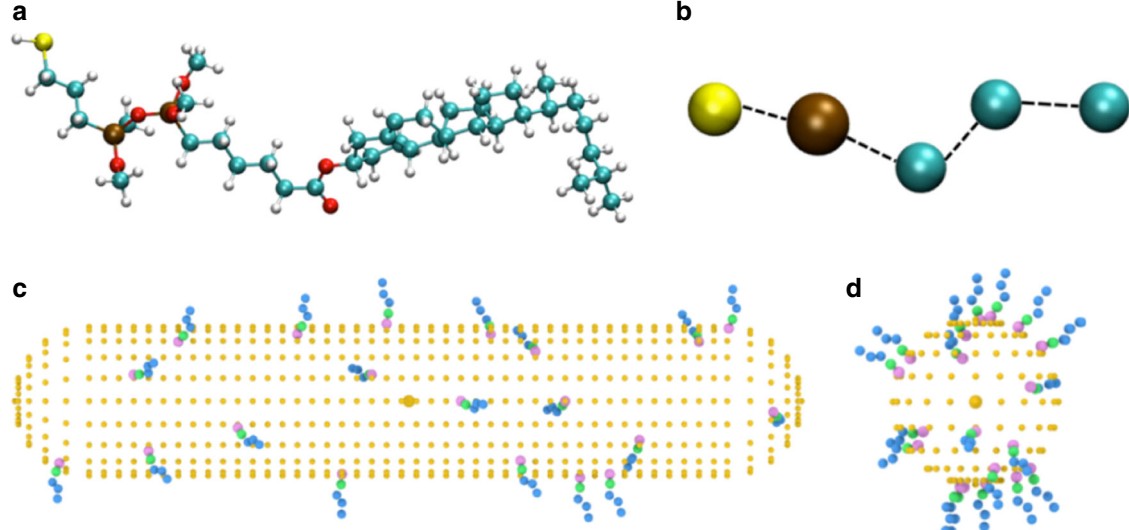

**Fig. 8** Atomistic and coarse-grained representation of the Chol-silane molecule. **a** Full representation of a chiral ligand molecule. **b** Coarse-grained description of the same ligand molecule, where only 5 of the original 118 atoms were retained: one Si, one S and three C atoms. **c** GNR1 spherocylindrical nanoparticle decorated with 20 CG Chol-silane ligands randomly distributed. **d** NP3 quasi-spherical Au nanoparticle bound to the same number of Chol-silane molecules in the Coarse Grained representation

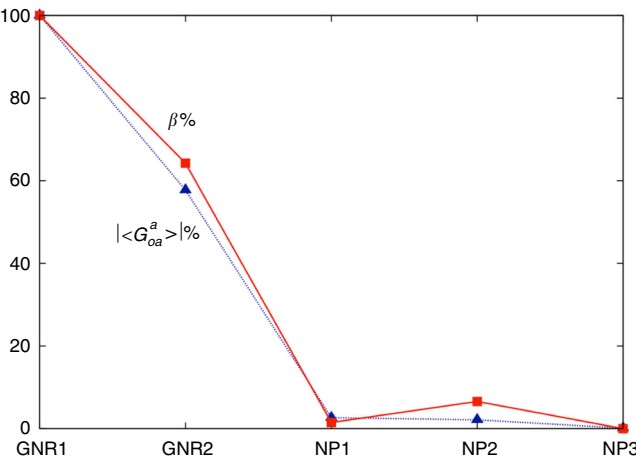

**Fig. 9** Trends of averaged chirality indicator $\left|\left\langle G_{oa}^{a}\right\rangle\right|$ for the Au NPs and the GNRs with 20 CG ligands. Values of chirality have been obtained for 20,000 random configurations on Au NPs and 1,800 selected configurations on GNRs. The cholesterol ligand molecules are bound to the Au NPs at random positions and orientations, while every combination of selected positions/orientations is analyzed for systems with GNRs

plotted these experimental data against the calculation results, converted into percentages as well, considering only systems with the highest number of CG Chol-silane ligands.

We see that the trend of the computed chirality index is in very good agreement with the experimental one. In particular the variation of chirality with size and shape is reproduced, without any fitting parameter. Noting that no specific material parameter for the nature of the GNRs or NPs has been introduced, these results seem to suggest that the origin of the chiral amplification effect could be connected to the fact that the chiral ligands are attached to a network, which makes them act together in a correlated way, augmenting, as we have seen experimentally, the overall chirality. This appears to support the firecracker ladder model considering a high local chirality of an ensemble of chiral ligands on the GNR surface that is capable of carrying a helical distortion over larger distances.

## Discussion

We presented experimental evidence that chirality, or more precisely chirality transfer, at the nanoscale can be significantly amplified by desymmetrization and shape complementarity of plasmonic nanoscale chiral additives in soft condensed matter. Our argument relies on a comparison of the actual number of chiral additive molecules in a given volume of an induced N*-LC continuum. A one order of magnitude lower number of chiral, here cholesterol, molecules induces an ever tighter helical pitch as we trade the neat organic molecules for ligand shells first on plasmonic Au NPs and then on GNRs. Anisometric chiral structures and assemblies ranging from a few nanometers to several tens of microns are omnipresent in nature, from DNA and collagen fibrils to the aforementioned narwhal tusks and plants tendrils. They are either responsible for, or the final product of, the amplification of chirality. The mechanism of chirality amplification through space, supported by calculations of a pseudoscalar chirality indicator, by shape desymmetrization combined with a cooperative effect of elastically-coupled feedback by helically assembled GNRs described here add additional tools for the exploration of amplification of chirality. We have demonstrated that one ubiquitous state of matter can be recruited to better understand the amplification of another ubiquitous

property of life-chirality. In addition, the simplicity of the system studied here will provide significant stimulus for exploring related chiral soft-nano material combinations for studies on dynamic chiral nanomaterial self-assembly[46], chiral liquid crystal-based skyrmions[47], chiral responsive metamaterials[48], and generally on chiral nanomaterials as catalysts in asymmetric syntheses[49] or employed as substrates for chiral discrimination and separation[50].

## Methods

**Materials**. Chemical precursors used for the synthesis were purchased from commercial sources (as indicated) and used without further purification (unless otherwise specified). Solvents were purified using a PureSolv solvent purification system (Innovative Technology Inc.). Hydrogen tetrachloroaurate ($HAuCl_4$) and ascorbic acid were purchased from Alfa-Aesar. Cholesterol, 4-(dimethylamino) pyridine (DMAP), dicyclohexylcarbodiimide (DCC), 6-bromohexenoic acid, sodium sulfate, sodium, (3-mercaptopropyl)trimethoxysilane, sodium hydroxide, cetyltrimethylammonium bromide (CTAB), chlorotrimethylsilane, sodium borohydride, silver nitrate, hydrochloric acid, and platinum(0)-1,3-divinyl-1,1,3,3-tetramethyldisiloxane (aka Karstedt's catalyst) were purchased from Sigma Aldrich. Deionized (DI) water, (Millipore, resistivity 18.1 MΩ cm) was used. 5CB was purchased from TCI America. PI-2555 polyimide was purchased from HD MicroSystems. Spherical spacers with different diameters were obtained from Dana Enterprises International, Inc. and Mylar film spacers were purchased from Artus Corporation. The chiral additive COC was obtained from Sigma Aldrich, R811, ZLI-4572 and CB-15 from Merck.

**Synthesis**. Syntheses and all spectroscopic characterization data are provided in Supplementary Note 1.

**Materials characterization**. Transmission electron microscopy (TEM) analysis was performed with a FEI Tecnai TF20 TEM instrument at an accelerating voltage of 200 kV. Samples were prepared by evaporating a drop of dilute GNR solutions in chloroform onto carbon-coated copper TEM grids (400 mesh, TED PELLA, Inc.), which were allowed to dry for 24 h prior to imaging. Freeze-fracture TEM (FF-TEM) images were obtained either on a FEI Tecnai TF30 ST TEM instrument at an accelerating voltage of 300 kV or a JEOL JEM-100S at 100 kV. The FF-TEM samples, replicas of fractured surfaces of the LC-GNR composites, were prepared following a procedure described elsewhere[51]. UV-Vis absorption and solution circular dichroism (CD) spectropolarimetry measurements were done using an OLIS spectrophotometer (1 cm path length quartz cuvettes). $^1$H NMR spectra were recorded in $CDCl_3$ on a Bruker DMX 400 MHz spectrometer and referenced internally to residual solvent peaks at 7.26 ppm. Polarized optical microscopy (POM) observations of the induced chiral nematic liquid crystals (N*-LCs) were recorded and photographed using an Olympus BX-53 polarizing microscope equipped with a Linkam LTS420E heating/cooling stage. Differential scanning calorimetry (DSC) was performed using a Pyris 1 DSC instrument (Perkin Elmer). Thermogravimetric analysis (TGA) was performed using a TGA Q500 (TA Instruments).

**Calculations**. Two different methods were used to analyze the GNR dimensions (aspect ratio) from the TEM images and calculate the total composition of the GNRs including ligand coverage. Method 1 made use of ImageJ[52], and at least one hundred GNRs were individually measured and the ligand coverage calculated taking into account the curvature perpendicular to the long axis of the GNR. Method 2 used an algorithm established in MATLAB that included both image analysis and full compositional calculation based on pure geometric considerations (see Supplementary Note 4).

**Sample preparation**. Admixing of the GNRs and other chiral dopants was achieved following standard protocol. Precise quantities were weighted into rigorously cleaned glass-vials using an ultramicrobalance. Standardized solutions of each chiral additive and N-LC host in a common organic solvent (purified $CHCl_3$) were prepared and the desired volumes of each solution were combined using calibrated Eppendorf pipettes and thoroughly mixed. Thereafter, the solvent was evaporated under mild vacuum. Homogeneous dispersions were ensured by mild, pulsed sonication at 20 °C for 5CB (at about 60 °C for Felix-2900–03), that is, with the mixtures in the induced N*-LC phase.

**Handedness of N*-LC**. Using an N*-LC with a known helical twist direction such as cholesteryl oleyl carbonate (COC, left-handed) we determined the twist direction of the N*-LC phase induced by GNR1 and GNR2 in 5CB using contact preparations. The absence of a discontinuation, that is, when two left-handed N*-LC phases make contact and a chiral N*-LC phase with a gradient pitch is observed in the contact zone, indicates that the handedness of each N*-LC phase is identical. A discontinuation between the fingerprint textures of two oppositely twisted N*-LC phases in the contact zone is characterized by the formation of a Schlieren texture typical for an achiral N-LC phase, which indicates opposite handedness.

**Induced circular dichroism (ICD) spectropolarimetry**. ICD measurements were performed using an OLIS Cary spectrophotometer. Thin films for ICD spectropolarimetry were prepared between two quartz substrates separated by ~ 10-µm Kapton tape spacers. Samples were rotated in 45° intervals from 0° to 315° (in the plane normal to the light beam) in order to differentiate CD absorption (reflection) from linear dichroism and birefringence as described elsewhere in more detail[53].

**Helical pitch measurements**. To measure $p$ of the induced N*-LC phases Cano wedge cells were used. Wedge cells are fabricated using two pre-cleaned glass substrates, on which polyimide PI2555 is spin-coated, baked, and rubbed to give strong planar anchoring. The substrates are then assembled unidirectionally such that there is a single Mylar spacer at one side of the wedge and no spacer on the other side. Epoxy glue was used as an adhesive at the edge of the cell, leaving three gaps for filling the cell. Both thickness variation and position were measured with an Ocean Optics spectrometer USB4000, where the variation in thickness with position was determined by each measurement individually (interference fringe method)[54]. Cells were then filled with the mixtures by capillary force, and then placed between crossed sheet polarizers to observe the disclination (Cano) lines with respect to position. All measurements were done at 20 °C for 5CB and 55 °C for Felix-2900–03. Additional methods used to determine $p$ measured the distances between either birefringent or dark domains in either homeotropic cells or cells with a free surface (air as top surface). Fingerprint textures were obtained in homeotropic cells constructed using two cleaned glass substrates spin coated with polyimide SE1211, and baked to provide strong homeotropic anchoring. Cells were assembled with a cell gap of 20 µm. After filling the cells with the N-LC/GNR mixtures by capillary force, they were sealed with epoxy adhesive. We used a microruler (calibrated with the microscope image capturing software QCapture Pro 6.0) to measure the distances between dark stripes. Free surface preparations employed pre-cleaned, but otherwise untreated glass substrates favoring degenerate planar anchoring, onto which one drop of the N-LC/GNR mixtures was placed. The top boundary was air, which favors homeotropic anchoring. Again, the distances between dark striations under crossed polarizers were measured using the microruler calibrated microscope software.

**Particle chirality quantification**. Details of chirality quantification are reported in Supplementary Note 14.

## Data availability

The authors declare that the data supporting the findings of this study are available within the paper and its supplementary information file. Any additional information that supports the findings of this study is available from the corresponding author upon reasonable request.

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

## Acknowledgements

This work was supported by the U.S. National Science Foundation (NSF, DMR-1506018), the Ohio Third Frontier (OTF) program for Ohio Research Scholars "Research Cluster on Surfaces in Advanced Materials" (T.H.), which also supports the Liquid Crystal Characterization facility at the Liquid Crystal Institute (Kent State University), where current TEM data were acquired and FF-TEM samples prepared. We would like to thank T. Turiv and O.D. Lavrentovich for their initial assistance in preparing some of test cells, the Swagelok Center at Case Western Reserve University for access to high-resolution TEM, and the TEM facility at NEOMED for further access to their imaging facility. Finally, L.Q. and C.Z. thank Italian MIUR: PRIN project 2015XJA9NT for financial support.

## Author contributions

T.H. and A.N. conceived the experiment. A.N. performed the syntheses and characterization of the materials. A.N. and L.L. performed TEM imaging. M.G. and A.N. performed FF-TEM imaging. A.N. and S.S. performed the POM experiments. A.N., S.S. and T.M. performed the calculations. L.Q. and C.Z. performed the chirality quantification calculations. T.H. and A.N. wrote the manuscript with contributions from all coauthors. T.H. directed the research.

## Additional information

**Competing interests:** The authors declare no competing interests.

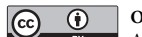

