## [Peer Review File · Nature Communications]

Reviewers' comments:

Reviewer #1 (Remarks to the Author):

The work by Nemati et al., describes the chirality transfer from gold nanorods capped with a network of chiral cholesterol molecules to achiral nematic liquid crystal host. The major highlight of the paper is that a far lower number of chiral molecules can induce helical distortion when bound on to gold nanorod surface. This is in fact a demonstration of remarkable chiral amplification induced by a tighter helical distortion in the field of the surrounding host liquid crystal molecules. However, it must be noted that the same approach has been earlier used for chiral amplification in similar liquid crystal host molecules using spherical gold nanoparticles as well as gold nanoclusters (ACS Nano, 2014, 8, 11966; ACS Nano 2016, 10, 1552; and few more). Parameters such as helical pitch and helical twisting power used in earlier publications are employed here as well, to establish the efficiency of chirality transfer. The additional input provided by this work is on investigating the aspect of shape anisotropy using gold nanorods as the guest substrate. The experiments have been well executed and the observed chiral effects are remarkably high. However, a clear understanding on the mechanistic details on the role of anisotropic structures would provide better insight that can attract general readership. For example, the authors in one of their similar publications stated that the small nanoparticles outperformed the bigger ones in terms of chiral induction. In contrast, the opposite effect is observed in the present case. Therefore,

(i) mechanistic details on the contribution due to the one dimensional nature of the nanorods are worth explaining in detail.

(ii) Less data is available on how the nanoparticles align inside the host material. Efforts can be made to check the images of Au nanoparticles and nanorods in the hybrids to draw a comparative analysis.

(iii) Also, comparison of anisotropic factors or other chiral parameters can provide newer dimensions to the work.

In summary, the high chiral induction effect reported merits publication if a clear physical insight into the mechanistic details is provided.

Reviewer #2 (Remarks to the Author):

Being familiar with chirality in liquid crystal systems, I have read the manuscript of Hegmann et al. with great interest. The manuscript describes how nanorods covered with chiral dopants are more efficient in transducing chiral information, compared to chiral dopants that would be dissolved in an (achiral) nematic liquid crystal, or even compared to nanoparticles that are functionalized with chiral dopants covalently. The authors attribute this effectiveness in chiral amplification to the shape anisotropy of the nanorods, and they suggest that the mechanism by which amplification occurs, is comparable to a firecracker ladder: the ligand-capped nanorods facilitate the twisting of the cholesteric helix.

The results are novel and original, and the work comes timely with an increasing interest of the condensed matter community with mechanisms by which molecular chirality can be amplified through space and across length scales (as outlined in the recent review of Fletcher et al, in Nature Nanotechnology). However, I have a few specific concerns.

I am not competent to judge the modelization efforts, but the experiments carried out by the authors are conclusive. In particular, Figure 5 is convincing, in the sense that the differences between helical twisting powers are large. On the other hand, I find the results described in Figure 6 a bit less convincing. In this figure the authors come up with the firecracker ladder analogy, which is appealing and interesting. However, to my knowledge, it is far from being proven as a general rule that any nanorods orient preferentially under the effect of the liquid crystalline environment – I am aware of some works by Smalyukh, such as Nano Lett. 2010, 10, 1347–1353,

that is cited in the text, in which direct experimental evidence was limited, and *Adv. Mater.* 2014, 26, 7178–7184. Fig. 6b and 6c suggest that the rods follow the helical organization of the cholesteric liquid crystal, but the images are very zoomed in, and they hardly prove that the orientation of the nanorods follows (or is influenced by) the cholesteric helix, e.g. that there is a twist among the nanorods. I feel that an unequivocal demonstration that the rods follow the orientation of the liquid crystal would support the proposed mechanism more convincingly. Maybe an analysis of larger area SEM or (cryo)-fracture-TEM images would help, and they could also contribute to the demonstration that there is only limited aggregation of the particles (currently the lack of significant aggregation remains an educated guess).

Another issue I have with the manuscript is its long introduction that pertains to the origin of homochirality in living systems. Indeed amplification and transmission of chirality are somewhat related to research on the origin of homochirality, but only from very far. The results reported by Hegmann et al. are interesting for what they are: an investigation of amplification of chirality in liquid crystals, mediated by nanoparticle anisometry. The link to the origin of life is wrong and misleading, and in my opinion must be removed entirely from both the abstract and the first paragraph of introduction – it is quite clear that the current work, while being undoubtedly interesting, will not contribute “to elucidate[ing] the origin of biological homochirality”.

The work of Soai and Blackmond is only very marginally relevant to the present work. Instead, there has been a lot of exciting studies related to amplifying, controlling and/or manipulating chirality in liquid crystal systems, and little of that work is currently cited in the paper. A short overview of what are the open questions in that research field, seems more appropriate to introduce the results described in this manuscript, for example (but not exclusively):

- Amplification of chirality in liquid crystals, by Eelkema and Feringa, *OBC* 2006
- Chiral Nematic Phase of Suspensions of Rodlike Viruses, by Grelet et al, *PRL* 2006
- Entropy-driven formation of chiral nematic phases by computer simulations, by Dussi and Dijkstra, *Nature Comm* 2016.
- Hierarchical Propagation of Chirality through Reversible Polymerization, by Ferrarini et al, *ACS Macro Letters* 2016.
- Chiral self-assembly of helical particles, by Giacometti et al, *Faraday Discuss.* 186, 171 - 186 (2016).
- Revolving supramolecular chiral structures powered by light in nanomotor-doped liquid crystals, by Brasselet et al., *Nature Nanotechnology* 2018

Minor comment: some expressions such as “some exceptional data that lucidly prove”, “a reasonably narrow size and shape”, or “almost exactly twice” need to be reformulated.

Reviewer #3 (Remarks to the Author):

The paper entitled 'Amplification of chirality by desymmetrization of chiral ligand-capped nanoparticles to nanorods probed and quantified in soft condensed matter' is demonstrating an enhancement of helical twisting power (HTP) in a chiral nematic (cholesteric) liquid crystal doped with chiral-ligand-decorated gold nano-rods (GNR). Although the enhanced HTP is known in gold nano-particles decorated with chiral ligands, the authors extended this idea further into GRDs and found that GRDs are much effective to the chiral induction as observed by HTP. I think that the work itself is interesting and well matured. The experimental data in beta and p values are reliable, and the control experiments are adequate. So I must appreciate the effort by the authors. However, still I am not really sure that the amplification and transfer of chirality, which the authors claim, are clearly explained in the paper. As the authors mention in the introduction part, transfer and amplification of chirality in the self-organization is very important. The point would be, (1) HTP is enhanced compared to the previous systems. (2) Helical twisting of GNRs is observed. (3) The elastic connection of the dispersed GNRs surely exists. However, although the phenomenon shown in this paper is important, the knowledge given in this paper is still very

limited and specific case observed in the helical liquid crystal. Is it possible to get insight into the general chirality in nature from the facts described in the paper? In my opinion, the paper is interesting enough for publication in materials science journals with a high impact factor, such as *Advanced Materials*, but not suitable for publication which requires the broad readership like in *Nature Communications*. Basically I like this work, but I am sorry not to support the authors.

Point-by-point response (blue font) to the referees' comments

Reviewer #1 (Remarks to the Author):

The work by Nemati et al., describes the chirality transfer from gold nanorods capped with a network of chiral cholesterol molecules to achiral nematic liquid crystal host. The major highlight of the paper is that a far lower number of chiral molecules can induce helical distortion when bound on to gold nanorod surface. This is in fact a demonstration of remarkable chiral amplification induced by a tighter helical distortion in the field of the surrounding host liquid crystal molecules. However, it must be noted that the same approach has been earlier used for chiral amplification in similar liquid crystal host molecules using spherical gold nanoparticles as well as gold nanoclusters (ACS Nano, 2014, 8, 11966; ACS Nano 2016, 10, 1552; and few more). Parameters such as helical pitch and helical twisting power used in earlier publications are employed here as well, to establish the efficiency of chirality transfer. The additional input provided by this work is on investigating the aspect of shape anisotropy using gold nanorods as the guest substrate.

It is indeed correct that we have performed earlier work on quasi-spherical, polyhedral nanoparticles as chiral inducers. However, the tremendous amplification of chirality reported here is significant and now supported by coarse-grained method calculation by one of the world's experts, Claudio Zannoni, that assume only geometric information.

The experiments have been well executed and the observed chiral effects are remarkably high. However, a clear understanding on the mechanistic details on the role of anisotropic structures would provide better insight that can attract general readership.

I am convinced that the additional FF-TEM data and the calculations of the chirality indicator provide these better understanding and mechanistic details the reviewer was asking for.

For example, the authors in one of their similar publications stated that the small nanoparticles outperformed the bigger ones in terms of chiral induction. In contrast, the opposite effect is observed in the present case. Therefore,

(i) mechanistic details on the contribution due to the one dimensional nature of the nanorods are worth explaining in detail.

The current study is significantly more detailed and we have learned a great deal by carefully calculating the molar helical twisting power and performing additional calculations. In fact, the next step will be to further adjust shape and size in conjunction with calculations to see how far we can push this concept.

(ii) Less data is available on how the nanoparticles align inside the host material. Efforts can be made to check the images of Au nanoparticles and nanorods in the hybrids to draw a comparative analysis.

The additional FF-TEM images continue to support that (i) the GNRs and the NPs are well dispersed and spaced according to our rather simple calculation of D_{p-p} . We would like to highlight that these FF-TEM experiments are by no means trivial and require enormous patience and experience from the experimenter. We have also prepared well-aligned samples that needed to be very small and required building a new cracking device from scratch. Unfortunately, due to the sandwiched nature, the presence of alignment layers (too many interfaces), we were not able to obtain a sample that could be imaged by TEM. To give you an idea of the aligned cell dimensions, below is a picture of what could be the smallest lab-made planar aligned cell.

Liquid Crystal Institute
Chemical Physics Interdisciplinary Program
P.O. Box 5190 • Kent, Ohio 44242-0001

330-672-2654 • Telefax: 330-672-2796 • E-mail: mail@lci.kent.edu • www.lci.kent.edu

(iii) Also, comparison of anisotropic factors or other chiral parameters can provide newer dimensions to the work.

In summary, the high chiral induction effect reported merits publication if a clear physical insight into the mechanistic details is provided.

We trust that the additional FF-TEM images and the calculations performed by our collaborator Prof. Zannoni do provide that extra level of detail and mechanistic insight the reviewer requested.

Reviewer #2 (Remarks to the Author):

Being familiar with chirality in liquid crystal systems, I have read the manuscript of Hegmann et al. with great interest. The manuscript describes how nanorods covered with chiral dopants are more efficient in transducing chiral information, compared to chiral dopants that would be dissolved in an (achiral) nematic liquid crystal, or even compared to nanoparticles that are functionalized with chiral dopants covalently. The authors attribute this effectiveness in chiral amplification to the shape anisotropy of the nanorods, and they suggest that the mechanism by which amplification occurs, is comparable to a firecracker ladder: the ligand-capped nanorods facilitate the twisting of the cholesteric helix.

The results are novel and original, and the work comes timely with an increasing interest of the condensed matter community with mechanisms by which molecular chirality can be amplified through space and across length scales (as outlined in the recent review of Fletcher et al, in Nature Nanotechnology). However, I have a few specific concerns.

I am not competent to judge the modelization efforts, but the experiments carried out by the authors are conclusive. In particular, Figure 5 is convincing, in the sense that the differences between helical twisting powers are large. On the other hand, I find the results described in Figure 6 a bit less convincing. In this figure the authors come up with the firecracker ladder analogy, which is appealing and interesting. However, to my knowledge, it is far from being proven as a general rule that any nanorods orient preferentially under the effect of the liquid crystalline environment – I am aware of some works by Smalyukh, such as Nano Lett. 2010, 10, 1347–1353, that is cited in the text, in which direct experimental evidence was limited, and Adv.

Liquid Crystal Institute
Chemical Physics Interdisciplinary Program
P.O. Box 5190 • Kent, Ohio 44242-0001

330-672-2654 • Telefax: 330-672-2796 • E-mail: mail@lci.kent.edu • www.lci.kent.edu

Mater. 2014, 26, 7178–7184. Fig. 6b and 6c suggest that the rods follow the helical organization of the cholesteric liquid crystal, but the images are very zoomed in, and they hardly prove that the orientation of the nanorods follows (or is influenced by) the cholesteric helix, e.g. that there is a twist among the nanorods. I feel that an unequivocal demonstration that the rods follow the orientation of the liquid crystal would support the proposed mechanism more convincingly. Maybe an analysis of larger area SEM or (cryo)-fracture-TEM images would help, and they could also contribute to the demonstration that there is only limited aggregation of the particles (currently the lack of significant aggregation remains an educated guess).

As requested by this reviewer, we have performed significant additional FF-TEM experiments and provided new images (new samples, zoomed out, even tried aligned samples – see comment to reviewer 1 above, multiple TEM instruments, etc.). Of the several hundred images, we provided new representative images that all support our earlier crude images. We do not see significant aggregation even in very zoomed out sample (see Supporting Information). In addition, we have worked with our collaborator, Claudio Zannoni, who performed additional calculations that support our funding perfectly, even the trends for the aspect ratio of the GNRs and the size of the Au NPs.

Another issue I have with the manuscript is its long introduction that pertains to the origin of homochirality in living systems. Indeed amplification and transmission of chirality are somewhat related to research on the origin of homochirality, but only from very far. The results reported by Hegmann et al. are interesting for what they are: an investigation of amplification of chirality in liquid crystals, mediated by nanoparticle anisometry. The link to the origin of life is wrong and misleading, and in my opinion must be removed entirely from both the abstract and the first paragraph of introduction – it is quite clear that the current work, while being undoubtedly interesting, will not contribute “to elucidate[ing] the origin of biological homochirality”.

We have changed the introduction and toned down the “origin of life” discussion significantly. While the examples were carefully chosen from chiral amplification events on surfaces, even nanosized or –structured, we have removed these section and focused instead, as suggested by the reviewer, on more specific studies related to chiral amplification in liquid crystals.

The work of Soai and Blackmond is only very marginally relevant to the present work. Instead, there has been a lot of exciting studies related to amplifying, controlling and/or manipulating chirality in liquid crystal systems, and little of that work is currently cited in the paper. A short overview of what are the open questions in that research field, seems more appropriate to introduce the results described in this manuscript, for example (but not exclusively):

- Amplification of chirality in liquid crystals, by Eelkema and Feringa, OBC 2006
- Chiral Nematic Phase of Suspensions of Rodlike Viruses, by Grelet et al, PRL 2006
- Entropy-driven formation of chiral nematic phases by computer simulations, by Dussi and Dijkstra, Nature Comm 2016.
- Hierarchical Propagation of Chirality through Reversible Polymerization, by Ferrarini et al, ACS Macro Letters 2016.
- Chiral self-assembly of helical particles, by Giacometti et al, Faraday Discuss. 186, 171 - 186 (2016).
- Revolving supramolecular chiral structures powered by light in nanomotor-doped liquid crystals, by Brasselet et al., Nature Nanotechnology 2018

All of these studies are now mentioned and briefly discussed and should serve as a more relevant introduction to the paper and the data presented.

Minor comment: some expressions such as “some exceptional data that lucidly prove”, “a reasonably narrow size and shape”, or “almost exactly twice” need to be reformulated.

We have made these corrections.

Liquid Crystal Institute
Chemical Physics Interdisciplinary Program
P.O. Box 5190 • Kent, Ohio 44242-0001

330-672-2654 • Telefax: 330-672-2796 • E-mail: mail@lci.kent.edu • www.lci.kent.edu

Reviewer #3 (Remarks to the Author):

The paper entitled 'Amplification of chirality by desymmetrization of chiral ligand-capped nanoparticles to nanorods probed and quantified in soft condensed matter' is demonstrating an enhancement of helical twisting power (HTP) in a chiral nematic (cholesteric) liquid crystal doped with chiral-ligand-decorated gold nano-rods (GNR). Although the enhanced HTP is known in gold nano-particles decorated with chiral ligands, the authors extended this idea further into GRDs and found that GRDs are much effective to the chiral induction as observed by HTP. I think that the work itself is interesting and well matured. The experimental data in beta and p values are reliable, and the controll experiments are adequate. So I must appreciate the effort by the authors. However, still I am not really sure that the amplification and transfer of chirality, which the authors claim, are clearly explained in the paper. As the authors mention in the introduction part, transfer and amplification of chirality in the self-organization is very important. The point would be, (1) HTP is enhanced compared to the previous systems. (2) Helical twisting of GNRs is observed. (3) The elastic connection of the dispersed GNRs surely exists. However, although the phenomenon shown in this paper is important, the knowledge given in this paper is still very limited and specific case observed in the helical liquid crystal. Is it possible to get insight into the general chirality in nature from the facts described in the paper? In my opinion, the paper is interesting enough for publication in materials science journals with a high impact factor, such as *Advanced Materials*, but not suitable for publication which requires the broad readership like in *Nature Communications*. Basically I like this work, but I am sorry not to support the authors.

We thank this reviewer for his critical statement, and we are convinced that the additional FF-TEM data and the calculations of the chirality indicator now provide that extra level of mechanistic understanding that makes this submission suitable for *Nature Communications*. The almost perfect fit of calculation and experiment ensure that future studies in this area are guided by both solid experimental data and calculations that only use geometric information to highlight that shape, size and especially aspect ratio of chiral nanoscale additives are key parameters in the chirality transfer and amplification in liquid crystal phases.

Liquid Crystal Institute
Chemical Physics Interdisciplinary Program
P.O. Box 5190 • Kent, Ohio 44242-0001

330-672-2654 • Telefax: 330-672-2796 • E-mail: mail@lci.kent.edu • www.lci.kent.edu

REVIEWERS' COMMENTS:

Reviewer #1 (Remarks to the Author):

The authors have addressed most concerns raised by the reviewers. The major concerns were related to the mechanistic details and the proof on the arrangement of nanorods inside the host materials. A better physical insight into the alignment of nanorods inside the host material is obtained through the additional FF-TEM images. Even though the images are not of very high quality to support the illustrated mechanism, this reviewer very well understands the experimental difficulty in obtaining images with a clear nanorod alignment in such hybrid materials. Therefore, I believe that the manuscript has substantially improved after the revisions and merits publication.

Reviewer #2 (Remarks to the Author):

This revised submission has answered rigorously and respectfully all the questions I raised on the previous version of the manuscript.

All points have been addressed, and also with additional theoretical contribution, the paper has improved significantly.

There is no doubt to this referee that the paper is worthy of publication, as it stands.

Reviewer #3 (Remarks to the Author):

The revised version of the manuscript "Amplification of chirality by desymmetrization of chiral ligand-capped nanoparticles to nanorods probed and quantified in soft condensed matter", including the newly-added calculation and freeze fracture TEM observation, looks much more attractive and scientifically sounds. Additional sentences in the introduction and the discussion parts are appropriate. Thus I would like to recommend the paper for publication in Nature Communications.

One minor comment;

Honestly, still I don't like the appearance of Fig. 1a. Why do the authors use multiple colors and shades for the same molecules?